# Heart Rate Variability Reflects Similar Cardiac Autonomic Function in Explosive and Aerobically Trained Athletes

**DOI:** 10.3390/ijerph182010669

**Published:** 2021-10-12

**Authors:** Alex Claiborne, Helaine Alessio, Eric Slattery, Michael Hughes, Edwin Barth, Ronald Cox

**Affiliations:** 1Department of Kinesiology, Nutrition and Health, Miami University, Oxford, OH 45056, USA; aclaiborne@bsu.edu (A.C.); slatteew@miamioh.edu (E.S.); barthef@miamioh.edu (E.B.); coxrh@miamioh.edu (R.C.); 2Department of Statistics, Miami University, Oxford, OH 45056, USA; hughesmr@miamioh.edu

**Keywords:** HRV, autonomic, aerobic, anaerobic, training

## Abstract

Autonomic cardiac function can be indirectly detected non-invasively by measuring the variation in microtiming of heart beats by a method known as heart rate variability (HRV). Aerobic training for sport is associated with reduced risk for some factors associated with cardiovascular diseases (CVD), but effects on autonomic function in different athlete types are less known. To compare cardiac autonomic modulation using a standard protocol and established CVD risk factors in highly trained intercollegiate athletes competing in aerobic, explosive, and cross-trained sports. A total of 176 college athletes were categorized in distinct sports as explosive (EA), aerobic (AA), or cross-trained (mixed) athletes. Eight different HRV measures obtained at rest were compared across training type and five health factors: systolic (SBP), diastolic blood pressure (DBP), body weight (BW), sex, and race. All athletic types shared favorable HRV measures that correlated with low CVD risk factors and indicated normal sympathovagal balance. A significant correlation was reported between DBP and pNN50 (% RR intervals > 50 ms) (*β* = −0.214, *p* = 0.011) and between BW and low-frequency (LF) power (*β* = 0.205, *p* = 0.006). Caucasian and African American athletes differed significantly (*p* < 0.05) with respect to four HRV variables: pNN50, HF power, LF power, and LF/HF ratios. Explosive, aerobic and mixed athletes had similar cardiovascular and autonomic HRV results in all eight HRV parameters measured. All athletes reported LF and pNN50 values that were significantly correlated with two CVD risk factors: DBP and BW. Compared with Caucasian teammates, African American athletes demonstrated lower LF/HF and higher pNN50, indicating an even more favorable resting sympathovagal activity and healthy CV function.

## 1. Introduction

Exercise training is integral to the maintenance and improvement of cardiovascular function, with high aerobic capacity predicting low risk of premature all-cause mortality [1]. Healthy cardiovascular adaptations typically found in aerobically trained athletes include increased oxygen pulse [2], increased stroke volume and reduced blood viscosity [3,4], resulting in increased cardiac output and high aerobic capacity (VO_2_max). Some studies suggest that these adaptations result in improvements in autonomic nervous modulation [5,6,7,8,9,10] which influence cardiovascular health. Improvements in health-related fitness can manage or even reverse cardiovascular disease pathology. Strength- and power-trained athletes differ from aerobically trained athletes in that they experience hypertrophy of the left ventricle wall [3], increased muscle size, and efficiency in torque and motor unit recruitment patterns [11]. Despite the expected health effects associated with intensive exercise training of either or both sport type(s), the development of cardiovascular disease in athletes is still possible. Cardiovascular disease infiltrates the health of intercollegiate student athletes, particularly in a few strength and power-related sports. A report on NCAA Division III football players discovered that these athletes had significantly more CVD risk factors compared with age-matched controls due to a higher incidence of body mass index (BMI) ≥ 30, hypertension, and metabolic syndrome [12]. 

Autonomic control of heart rate has been discussed as a potential biomarker of CVD risk [13,14,15]. Autonomic function can be detected non-invasively by measuring small changes in timing of heart beats and reporting heart rate variability (HRV) (14–15). Time domain variation represents flexibility of the autonomic system to respond to blood pressure variability [16] in order to manage cardiac output. HRV also provides a display of resting sympathetic neural activity at the sinoatrial (SA) node via reporting high (HF)- and low (LF)-frequency domain indices [17]. The theoretical ratio of sympathetic (represented by LF) to parasympathetic (HF) modulation of HRV is increased in response to physical or psychological stress and returns to baseline when the stress is removed. The calculated LF/HF ratio at rest reflects sympathovagal balance, which represents resilience of the autonomic nervous system to stressors, and if compromised, reflected by an elevated LF/HF, can indicate disease [17]. Studies of HRV in diseased populations have reported that sick patients typically have low heart rate variability even at rest, which may be indicative of inappropriate cardiac reflexes [13,14,15]. It is clear that aerobic exercise training increases resting HRV and that higher-frequency and -intensity exercise provide greater return to resting homeostasis as indicated by sympathovagal balance, reflected by a lower LF/HF ratio [18]. In fact, well-trained athletes are prime examples of individuals with a lower LF/HF ratio at rest [19,20,21] given the strong vagal effect on the heart. Even in sedentary and old subjects who take on intense exercise training, some HRV variable improvements track with improvements in cardiovascular fitness and health [22,23,24]. While not yet defined, future criteria for ‘healthy’ HRV could indicate cardiac health in a non-invasive method more precise than simply measuring resting heart rate (HR) [15].

HRV has been studied at rest [7] and following various exercise training programs [25,26,27,28,29,30,31,32,33]. Results show advantageous shifts in autonomic control in aerobically trained individuals such as increases in resting time domain HRV parameters (e.g., root-mean-square of successive RR differences (RMSSD), standard deviation of RR intervals (SDNN) and percent of successive RR intervals > 50 ms (pNN50)). These deviations in RR intervals have been found to characterize healthy autonomic function [34,35,36,37]. It is of interest from performance and health aspects to measure time domain, non-linear and frequency parameters of HRV in a variety of athletes to understand how different training regimens may influence cardiac and autonomic nervous system regulation. This can occur by examining timing of RR intervals and high- and low-frequency oscillations of heart rate. In addition to other biomarkers of health (e.g., blood pressure, body weight, and body mass index), some HRV parameters may provide valuable insight on cardiac health and performance in young intercollegiate athletes training in aerobic, and especially in strength and power sport athletes, whose cardiovascular health is often overlooked.

Most previous HRV studies have focused on aerobic training, with physiological characteristics of endurance athletes directly correlating to many HRV factors. Strength and power athletes train differently and present different physiological characteristics. Since there are known differences in physiological effects between aerobic and strength training [38], the present study assessed HRV in athletes categorized into three training types: aerobic, explosive, and mixed. Eight HRV measures were compared with several established CVD risk factors in young, healthy college athletes. The current study explored time and frequency domain in apparently healthy college athletes that train differently in terms of strength, power, and endurance. It was hypothesized that aerobic athletes would present HRV variables reflecting parasympathetic over sympathetic nervous system influence (representing a more favorable sympathovagal balance) at rest compared with mixed and explosive athletes. Significant correlations were expected between HRV measures and most traditional CVD risk factors, with the assumption that higher HRV would correlate with lower CVD risk factors.

## 2. Methods

Sample and data collection. Participants (N = 176; 81 M, 95 F) in this study included both male and female Division I collegiate athletes with typical daily training sessions of at least 2 hr. A group of similar age, healthy and recreationally active control subjects were studied. All subjects were recruited using personal contact with coaches, Sports Medicine doctors, and contact through university classes. All participants met the following criteria prior to participation: 1. ages 18–25 years, 2. currently healthy without cardiopulmonary limitations, and 3. free of any injury preventing them from exercising or training at the time of this study. A health history questionnaire identified individuals with elevated risk; however, none reached the level of exclusion. Subject demographics are reported in Table 1. All procedures were approved by the Institutional Ethics and Review Board (IRB), and all subjects gave written informed consent before participating in this study.

Groups were formed to include explosive athletes (EA) if their typical sport performance during competition lasted <20 s bouts. Cross-trained athletes (mixed) performed in competitive bouts lasting 20 < *x* < 60 s. Aerobic athletes (AA) performed >60 s bouts during competition. African American males athletes (*n* = 10) participated in mostly power sports. Their participation was in short-distance track (100–200 m, 400 m, hurdles, *n* = 2), and football (RB/WR/Safety *n* = 4, line *n* = 4). In football, Caucasian males (*n* = 24) participated in line positions (*n* = 14), running back (RB) RB/wide receiver (WR)/Safety (*n* = 6), kicker (*n* = 2), and as tight ends (*n* = 2). A majority of Caucasian football athletes were linemen with higher body fat, while African American athletes were more frequently RB/WR/Safeties with lower body fat.

Other Caucasian male athletes (*n* = 32) participated in baseball (*n* = 8), short- (*n* = 6) and long-distance swim (*n* = 5), diving (*n* = 4), cheerleading (*n* = 1), short- (*n* = 4) and long-distance track (*n* = 2), and endurance cycling (*n* = 2). African American female athletes (*n* = 8) participated evenly among sport types: basketball (*n* = 4), volleyball (*n* = 1), and short-distance track (*n* = 3). Caucasian female athletes (*n* = 43) participated in field events (*n* = 4), short- (*n* = 7) and long-distance swim (*n* = 11), softball (*n* = 9), diving (*n* = 4), volleyball (*n* = 2), field hockey (*n* = 7, including 1 goalie), soccer (*n* = 10), marathon (*n* = 1), short-distance track (*n* = 1), triathlon (*n* = 1), basketball (*n* = 5), and ice skating (*n* = 3). 

### 2.1. Study Design 

Over a ten-month period, subjects reported to the Kinesiology laboratories between 07:00 and 10:00 h after an overnight fast (10–12 h), refraining from caffeine and exercise for at least 8 h, and maintaining their normal workout routine the previous day. For HRV testing, subjects were instructed to sit upright, quietly, and without any distractions (e.g., moving, talking, text messaging) for a period of 10 min, in a well-lit, quiet room, away from all other laboratory procedures. Two septal lead electrodes (CamNtech ActiWave Cardio) recorded ECG signals, and manual artifact correction was applied if artifact clouded data were detected. In a few cases, if the artifact was severe enough, but did not impede visualization of R wave peaks, a medium-grade artifact correction function intrinsic to Kubios HRV software was applied. *Kubios HRV 2.2* software (Kubios, Kuopio, Finland) identified R-wave peaks, and calculated SDNN (ms), RMSSD (ms), pNN50 (%), LF (m^2^), HF(m^2^), LF/HF Ratio, Poincare SD1 (ms) and SD2 (ms). A list of variable calculations is provided in Table 2.

A common concern from HRV studies is the need for standardization of recording criteria if HRV is used in fitness or cardiovascular risk assessment. Aubert and colleagues recommend a 10 min minimum tracing for HRV analysis [27,31]. 

This allows time for HRV to fluctuate from minute to minute and is naturally more representative of how HRV may vary over periods of rest. ECG recordings of longer length may increase validity, as there is a higher number of R-waves from which to calculate SDNN, pNN50 and RMSSD but there would be challenges with standardizing testing conditions over long time periods. For practical reasons, however, HRV recordings typically lasted 2–20 min. According to our pilot data, recordings of 10 min in a person seated quietly in a small room were reproducible. As HR increases, a decrease in SDNN will occur, so it is recommended to use standard deviations corrected by HR, or average RMSSD to correct for influence of HR on time domain. For these reasons, and following the recommendation by Aubert [27,31], we used 10 min of HRV data collection, which provided a reproducible measurement that resembled a normal resting heart rate.

Hydration status can affect HRV variables [39,40]; however, previous studies have reported dehydration occurred following exercise and exercise-induced heat stress. Since the athletes in this study all refrained from physical activity for 8 h and were in a 12 h fasted state prior to HRV testing, hydration status was not assumed to be a variable of interest on HRV measurement or body weight (BW). BW was measured by the Bod Pod scale (COSMED USA, Inc., Concord, CA, USA), calibrated to the nearest 1 g and height measured by a Health-O-Meter 209HR wall-mounted Stadiometer, (Global Equipment Company, Robbinsville, NJ, USA) with an accuracy of 0.1 mm. Thereafter, body mass index (BMI) was calculated. Body fat % (BF) was estimated from body density by air displacement in a Bod Pod. Systolic (SBP), diastolic (DBP), and mean (MAP) blood pressure were measured with an automated Omron Digital blood pressure monitor, HEM-907XL (Omron Healthcare, Kyoto, Japan) that measured pressure waves associated with Korotkoff sounds and was calibrated daily. HRV testing occurred after all other tests were completed.

### 2.2. Statistical Analysis

To investigate associations between HRV measurements and subject fitness characteristics, eight stepwise regression analyses were fit modeling each of the HRV measurements as response variables with age, fitness variables, and the three CVD variables as predictor variables. Standardized regression coefficients (*β*-weights) were used to assess the strength of association between a given HRV response and predictor, adjusted for other predictors in the model. *p* ≤ 0.05 was used as the selection criterion for variable inclusion. 

One-way ANOVAs were used to test for differences between athlete types with respect to each of the eight HRV variables. HRV variables that were right-skewed received appropriate power transformations prior to analysis to satisfy normality and variance homogeneity assumptions, and effect sizes (η^2^) were calculated.

Race and sex were investigated using two-way (2 × 2) ANOVA on each HRV variable. All standard ANOVA assumptions were checked and verified, with appropriate transformations applied to achieve normality and homogeneity of error variance.

All statistical analyses were performed using R [41]. Statistical differences were deemed significant at *p* ≤ 0.05.

## 3. Results

Twenty-one subjects met the American College of Cardiology criteria [40] for hypertension (SBP ≥ 130 mmHg), 46 were overweight (BMI ≥ 25), and 16 were obese (BMI ≥ 30). Stepwise regression models (Table 3) revealed that BW was significantly (*p* < 0.05) associated with HRV variables: LF (*β* = 0.205), HF (*β* = −0.207), and LF/HF (*β* = 0.208). DBP was associated with SDNN (*β* = −0.211), pNN50 (*β* = −0.214), RMSSD (*β* = −0.197), SD1 (*β* = −0.197), and SD2 (*β* = −0.199). BF was negatively associated with SDNN (*β* = −0.207), RMSSD (*β* = −0.160), SD1 (*β* = −0.160), and SD2 (*β* = −0.207). No other candidate predictors significantly associated with the HRV responses, and none of the associations that were statistically significant were very strong. Furthermore, none of the eight models exhibited strong predictive performance (max *R*^2^ = 0.087). 

While females seemed to generally exhibit slightly more variability in HRV responses, there was no evidence of systemic sex differences in mean responses. One-way ANOVAs failed to detect significant differences between athlete types and controls for all HRV variables with small effect sizes (η^2^ < 0.02). Violin plots display the distributions of LF (Figure 1), HF (Figure 2), RMSSD (Figure 3), and pNN50 (Figure 4) based on athlete type and sex.

Even with a small number of African American athletes in this study, a significant race effect was detected for pNN50, LF, HF and lnLF/HF. African American student athletes had significantly lower LF/HF ratios than Caucasian. Caucasian student athletes were found to have significantly higher LF than African American (F_(1, 136)_ = 5.20, *p* = 0.024), whereas African American student athletes were found to have significantly higher HF (F_(1, 136)_ = 5.17, *p* = 0.024) and pNN50 (F_(1, 136)_ = 5.74, *p* = 0.018) than Caucasian. No race by sex interactions were detected, nor was sex a significant factor in any of these analyses.

## 4. Discussion

This study addressed associations between select HRV parameters and cardiovascular disease risk factors in aerobically, power-, and mixed-trained athletes. Individuals playing strength- and explosive-type sports or positions that benefit from high BW and BMI presented with several CVD risk factors that were unexpectedly high. Significant associations were observed between % BF, BW, MAP and DBP. BF and DBP were negatively associated with time domain HRV (SDNN, RMSSD, and pNN50), while BW was positively associated with frequency domain HRV (LF, HF). A significant racial disparity in the athletes was unexpected; African American athletes showed significantly higher HF power, while Caucasian athletes displayed significantly higher LF power. A less favorable sympathovagal balance leaned more towards sympathetic neural input in Caucasian athletes, while a more favorable parasympathetically dominant occurred in African American athletes. 

With most HRV research still focused on aerobic athletes, further research is warranted to gain insight into CV health and autonomic function of athletes who play positions in sports that require short bursts of strength and power and who might have some CVD risk factors even at a young age. When measured at rest, aerobic athletes have typically presented lower LF power, higher HF power, higher RMSSD and pNN50 while standing and higher LF, SDNN, and RMSSD while supine compared with non-athletic controls [10]. Correlations were reported between actual VO_2_max and supine LF (*r =* −0.32, *p* = 0.04) and LF/HF (*r =* −0.32, *p* = 0.04) [42]. These results indicated that endurance-trained individuals likely had lower sympathetic activation of cardiac muscle when measured in the supine position. Significant correlations have also been found between VO_2_max, RMSSD (*r* = −0.30 *p* = 0.04), pNN50 (*r* = −0.30, *p* = 0.04), and Poincare SD1 (representing values of each pair of beat to beat intervals) (*r* = −0.30, *p* = 0.04) in moderately active college-aged males [25]. Resting autonomic activity appears to be under more vagal control in trained endurance runners than in healthy age-matched controls [9,10]. To assert the effect of training on HRV, one group compared effects of a one-year progressive endurance exercise program on HRV frequency domain, and although no significant effect on VO_2_ max was revealed, endurance exercise training yielded higher HF compared to the control group [20]. Other short-term high-intensity exercise programs have elicited a training effect on LF, HF, and LF/HF [23]. Studies tend to agree that both extended-duration moderate and short-duration intense exercise training can influence some parameters of HRV, implying a general and favorable effect of enhancing PNS over SNS influence on cardiac function during rest [22]. 

Thus far, only a few studies have compared HRV among aerobic and strength/explosive athletes. One study suggested that HRV can provide an early warning for overtraining in female wrestlers [43]. When comparing HRV and actual VO_2_max in aerobic-trained cyclists with weight lifters, a correlation has been reported between VO_2_max and SDNN [21]; greater variability was associated with athletes who had higher cardiorespiratory fitness, though both athlete types had similar 24 h HRV. A separate comparison of elite track and field athletes separated into endurance and power events found a difference in LF power between distance and sprint athletes [19]. Additionally, it seems that RMSSD responds differently in aerobic- and explosive-trained males. The different graphic shapes elucidate different distributions of HRV indexes among the athlete types, an intriguing point for future research to focus on. The index pNN50, referring to the percent of successive NN intervals varying greater than 50 ms from each other, suggested different distribution between aerobic-trained males and strength-trained males, but a larger sample size is needed for statistical power. There have not been extensive studies nor a consensus about HRV and power athletes from standardized research methodology; however, the current study shows that most resting HRV variables of cross-trained and power college-aged athletes are similar to those in aerobically trained college-aged athletes.

A surprising finding of this current study was the significant race differences. Compared with Caucasian Americans, African Americans experience a higher prevalence of cardiovascular diseases that present early in life [44] and occur at every age category across the life span [45,46]. The variables race and ethnicity have been shown to influence autonomic regulation of cardiac function. A study on HRV, race, ethnicity and class defined class by including education, occupation and income, and reported that higher social class was associated with higher values of all HRV parameters measured including LF, HF, and SDNN [47]. A study [48] compared HRV parameters in 403 black and 461 white healthy non-athletic adults, aged 20–30 years, living in South Africa. Significant differences were reported in SDNN between black (150.2 ± 43.8 ms) and white (158.6 ± 45.0 ms), RMSSD (black = 49.3 ± 21.2 ms, white = 46.1 ± 20.9 ms), LF (black = 58.6 ± 12.6 nu, white = 66.3 ± 11.4 nu), HF (black = 38.5 ± 11.8 nu, white = 31.6 ± 10.9 nu) and LF/HF ratio (black = 1.8 ± 1.0, white = 2.5 ± 1.4). The percent differences of HRV parameters showed a similar pattern for black and white subject comparisons in the current study that had a small sample size compared with the Kochli study [48] for frequency domain HR markers LF (23% vs. 12.3%) and HF (29.0% vs. 19.7%). Time domain markers, SDNN (8.3% vs. 5.4%) were similar comparing male data (8.3% vs. 6.7%) but the difference was greater due to higher female SDNN values in the current study (8.3% vs. 6.7%). Both Kochli and the present study reported significant HRV differences due to race.

In the current study of college athletes, African American athletes had a 49% lower LF/HF, indicating a more favorably balanced SNS:PNS activity at rest, as well as 60% higher pNN50, indicating a greater inter-heartbeat variability in consecutive resting sinus intervals, compared with their white teammates. The association of race and HRV could be influenced by particular race participation in sports, for example, in this study, all swim athletes were Caucasian and a significant number had high DBP and SBP. Additionally, a majority of football lineman athletes, with above average BW, were Caucasian. Discrepancies exist when comparing HRV by race and ethnicity, with one study reporting that compared to Caucasians, all HRV parameters in African Americans were lower, which implicated autonomic dysfunction [49]. Choi et al. [49] reported a decrease in PNS rather than an increase in SNS in young adult African Americans, which can result in hypertension over time. A recent systematic review suggests that African American individuals have higher vagal tone as represented in HF, compared with white or euro-Caucasian individuals [50]. In the present study, race was a powerful factor in our models, accounting for 3-foldthe effect of sex and 7-fold the interaction of race and sex. Results of this study support this possibility, with several key HRV parameters in African American athletes representing more favorable resting cardiac regulation compared to white athletes.

In the present study, a tendency toward greater HRV in females was observed in LF and HF (Figure 1 and Figure 2), but not RMSSD and PNN50% (Figure 3 and Figure 4). Across many HRV indices and among both races, violin plots indicated somewhat higher variability in HRV among females, but also suggest that a few outliers exist in females. Whether these outliers are examples of very high, yet still prognostically good autonomic function, or whether they are indicative of dysregulation of autonomic function warrants further investigation. A study comparing HRV modulation between genders reported different PNS and SNS based on LF and HF, with females having a greater vagal component and males having a greater sympathetic component at rest [51,52,53]. Another study supported this by reporting that indices of PNS such as HF, RMSSD and pNN50 were higher in women compared to men [50]. While our data also show a trend for more parasympathetic dominance in females compared to males, sex was not a significant factor for HRV.

Despite the inclusion of 176 college athletes representing 15 distinct types of sports, there were clear limitations in this study: (a) 82% of the athletes identified as Caucasian, (b) some sports had only one participant (e.g., marathon, triathlon, and cheerleading), (c) some athletes were competing in season, others were off-season, (d) participants were assumed to be hydrated, and (e) all were young, aged 18–25 years old. Therefore, the generalizability of the results is limited to young, fit college-aged athletes.

## 5. Conclusions

In a large controlled collegiate athlete cohort study that assessed eight HRV parameters over a standard 10 min resting test time in aerobic, power, and cross-training athletes, it was determined that training type had no effect on the HRV parameters measured, thus power training may elicit similar benefits on neuro-cardiac function as aerobic or mixed training. BW was positively associated with LF, indicating that athletes with higher body weight had higher resting sympathetic activity, which is typically not favorable. DBP was negatively associated with pNN50, indicating that the lower DB,P the higher the variability between NN intervals, which is typically favorable. Compared with Caucasian teammates, African American athletes demonstrated two significantly higher HRV parameters, lower LF/HF and higher pNN50, indicating a more favorable resting PNS:SNS balance. These results suggest that a 10 min HRV measurement was found to be a practical, reproducible, and valuable non-invasive screening tool for assessing cardiovascular health focusing on autonomic regulation and sympathovagal balance in young, athletic college-aged students.

## Figures and Tables

**Figure 1 ijerph-18-10669-f001:**
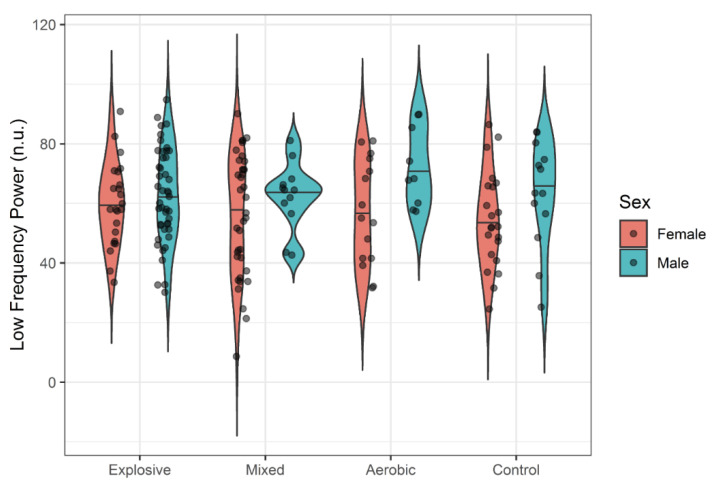
LF by Athlete Type.

**Figure 2 ijerph-18-10669-f002:**
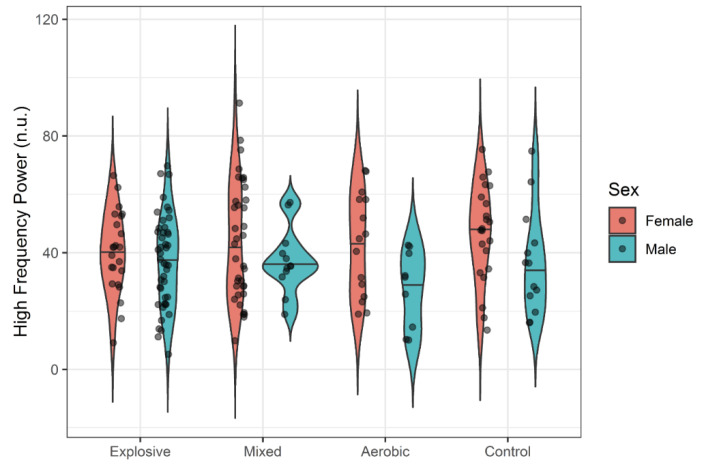
HF by Athlete Type.

**Figure 3 ijerph-18-10669-f003:**
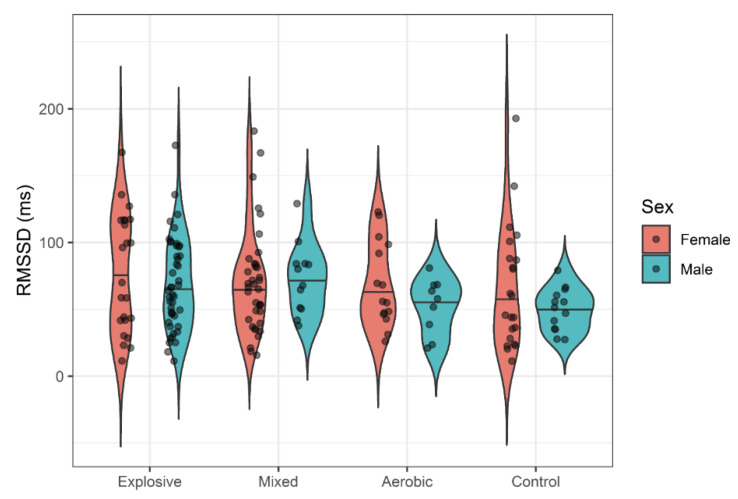
RMSSD by Athlete Type.

**Figure 4 ijerph-18-10669-f004:**
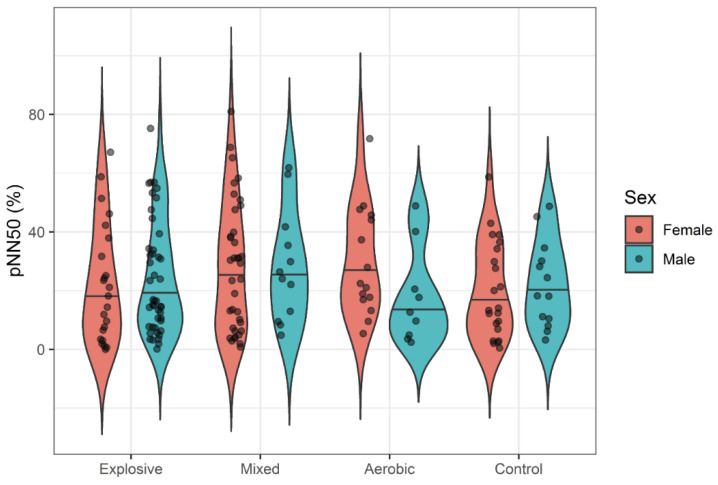
pNN50 by Athlete Type.

**Table 1 ijerph-18-10669-t001:** Subject demographics.

	Explosive (*n* = 68)	Mixed (*n* = 48)	Aerobic (*n* = 24)	Control (*n* = 36)
Age (years)	19.6 ± 1.2	19.4 ± 1.22	19.6 ± 1.34	20.2 ± 1.53
Sex	23F, 45M	36F, 12M	15F, 9M	21F, 15M
Race	55 white, 13 black	43 white, 5 black	24 white	36 white
Body Weight (kg)	88.73 ± 22.75	66.85 ± 8.94	69.06 ± 6.56	66.43 ± 11.79
Body Fat (%)	17.3 ± 17.36	18.9 ± 7.42	18.1 ± 7.29	
Height (cm)	179.18 ± 10.89	170.99 ± 9.49	173.6 ± 7.97	168.88 ± 7.61
Body Mass Index	27.31 ± 4.88	22.85 ± 1.97	22.92 ± 1.57	23.22 ± 3.45
Systolic BP (mmHg)	120 ± 13.75	109 ± 10.32	111 ± 12.32	
Diastolic BP (mmHg)	68 ± 10.1	63 ± 6.53	61 ± 7.5	
MAP (mmHg)	85	78	78	

**Table 2 ijerph-18-10669-t002:** Heart Rate Variability metrics.

HRV Metric	Units	Description
Heart rate	Beats/min	Frequency of depolarization of the sinoatrial node, stimulating systole
NN interval	ms	The varied duration of time between R waves in two successive heart beat QRS complexes
SDNN	ms	Standard deviation of time between R waves
pNN50	%	Percent of successive NN intervals varying greater than 50 ms in duration from each other
RMSSD	ms	Root mean square of the standard deviation of NN interval variation
LF	ms ∗ ms	Low-frequency power, a biomarker of sympathetic neural input
HF	ms ∗ ms	High-frequency power, a biomarker of parasympathetic neural input
LF/HF		The balance of LF and HF power, indicating balance of resting cardiac autonomic control between sympathetic and parasympathetic nervous systems
SD1	ms	In a Poincare plot, the longitudinal range of all NN intervals in a recording
SD2	ms	In a Poincare plot, the latitudinal range of each successive NN intervals in a recording

**Table 3 ijerph-18-10669-t003:** Standardized regression coefficients (*p*-values) for significant predictors in stepwise regression models.

	SDNN (ms)	RMSSD (ms)	pNN50 (%)	LF (nu)	HF (nu)	*ln* LF/HF	SD1 (ms)	SD2 (ms)
Body Weight (kg)	-	-	-	0.205 (0.006)	−0.207 (0.006)	0.208 (0.006)	-	-
Body Fat (%)	−0.207 (0.012)	−0.160 (0.052)	-	-	-	-	−0.160 (0.052)	−0.207 (0.012)
Body Mass Index	-	-	-	-	-	-	-	-
Systolic BP (mmHg)	-	-	-	-	-	-	-	-
Diastolic BP (mmHg)	−0.211 (0.010)	−0.197 (0.018)	−0.214 (0.011)	-	-	-	−0.197 (0.018)	−0.199 (0.016)
MAP (mmHg)	-	-	-	-	-	-	-	-
Age	-	-	-	-	-	-	-	-
Model *R*^2^	0.087	0.064	0.046	0.042	0.043	0.043	0.064	0.082
AIC	1347.2	1407.0	1239.7	1491.6	1492.7	420.8	1309.5	1408.7

## Data Availability

The data presented in this study are available on request from the corresponding author.

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
