# Peer review of "Heart Rate Variability Reflects Similar Cardiac Autonomic Function in Explosive and Aerobically Trained Athletes"

_ijerph, 2021, doi:10.3390/ijerph182010669_

Round 1

Reviewer 1 Report

The Purpose, ( "To compare cardiac autonomic modulation using a standard protocol and established cardiovascular disease (CVD) risk factors in highly trained intercollegiate athletes competing in aerobic, explosive, and cross-trained sports") it is interesting but the results are not balanced on this focus. I personally don’t like the use of the term “race effect”, in any case the “race effect” cannot be considered if 18 vs 122 athletes were compared.

Maybe, by deleting these issue (and table 4 and figure 5) from the paper,  a new submission can be considered.

Minor Revisions

Abstact

-A space i missinb before Introduction

Height HRV measure but after : only 5 are mentioned

Last row: LF:HF   maybe is  to use  always LF/HF  

Introduction

Line 3 a full stop is missing after [1]

-“can reflect certain health outcomes”  Is too much generic

-Rephrase and define PNS and SNS:

It was hypothesized that aerobic athletes would demonstrate HRV variables that favor PNS over SNS influence (representing a more favorable sympathovagal balance) at rest compared with mixed and explosive athletes.

Pag 2 last rows:

HRV measures have demonstrate that at rest PNS/SNS >1?

Please use Mixed and EA athletes or Cross-trained  and explosive athletes

methods

Please add numbers: Male (81) and Female (95)

Pag 4 please add SD1 e SD2 in table 2

-table 2 ms *ms ? please define power spectrum

-a ratio is adimensional  and his logarithm too, please delete ms in table 3, 4  and fig 5

“It is undeniable that race and class influence autonomic regulation of cardiac function”  It is not clear

Reviewer 2 Report

Many thanks for the exciting Study "Heart Rate Variability Reflects Similar Cardiac Autonomic Function in Explosive and Aerobically Trained Athletes", which I read with great interest. I have few comments, which will be incorporated into the publication. 

1. Please add a clear limitation sector of the study.

2. Why was not a 24h HRV measurement performed? Please argue your chosen measurement method based on previous studies in the discussion.

3. A surprising finding of this current study are the significant race differences.

You may add the following recent study to your discussion: Köchli et al. Adiposity and physical activity are related to heart rate variability: the African-PREDICT study https://pubmed.ncbi.nlm.nih.gov/32589287/

Minor revision:

Change LF:HF ratio to LF/HF ratio.

(as the ratio is also written as LF/HF in other publications)

Reviewer 3 Report

Dear Authors

You have written an interesting study. From my point of view the introduction is clearly written and it nicely leads to the main research question of your study.

The methods section is solid and also clearly written. However, please add the model of weight scale, stadiometer and blood pressure monitor. Also please clearly state if these measurements were done before every HRV measurements or just at the beginning of the study.

Conclusion: ''or assessing cardiovascular and autonomic function''  FOR WHAT??? The practical implications are poorly written. Try to present the practical application of your findings clearly and directly. Amend

Overall, the paper is sound and I recommend acceptance after minor revision.

Kind regards

Reviewer 4 Report

The paper  is  very  interesting . The research  is  well  conducted . Some  aspects fiundamental  for correct  interpretation of the  data  are missing , especially  for the hydration status  of the  athletes , that  can be  different  by gender  and also by etnicity and by the intensity  of training . 

The eventual  correlation with the BW could assume  an other impact . 

Round 2

Reviewer 1 Report

Thank you for addressing  my concerns in your last revision, but I'm not yet in agreement about the statistical analysis and the number of subjects considered in black and white comparison. I am still concerned about difference in race are not well supported or reported: considering the number of table 4, I observe that if the analysis was conduct on 10 vs 56 (male) and 8 vs 66 (female), black subjects are to much less than white subjects; even if you can assess  that no difference exist between male and female on the same sample (only first two groups of table 1),  even considering 18 vs 122, in my opinion black subjects are not enough.

Further some results are not clearly reported:

Why is it F (1, 136) ?  I can found this number 136 when I sum the subjects. What p the symbols (*) describe? all the column report it. Usually you explain what group is not different respect to....

I think that table 4 can be delete and the author can report statistical results about black and white differences.

Even if without race analysis you can conclude as reported:

"These results suggest that a 10 min HRV measurement was found to be a practical, reproducible, and valuable non-invasive screening tool for assessing cardiovascular health focusing on autonomic regulation and sympathovagal balance, in young, athletic college aged students."

Minor revisions:

  1. Delete space after minus  power (? _= 0.205, p= -0.214 and after mixed
  2. Rewrite this sentence "all athletes reported LF and pNN50 values that were significantly correlated with CVD risk factors: DBP and BW."
  3.  Introduction: Change 0700 and 1000 h in 07:00 and 10:00
  4. Remove explained from Table 2. Heart Rate Variability metrics explained. Change.  “N-N interval”.  With  “RR interval” in table 2 and in the text
  5. pag 9 How is reckoned or estimate VO2max ?
  6. Pag 10 It is not clear: "Although in that study gender was not compared"

(please find in yellow six the points and relative phrases)

Author Response

We agree to delete Table 4 and refer to differences in Black and white athletes only in the text. See highlighted sections on page 8

Regarding the query about the F value:

There are 56+66+10+8 = 140 subjects in the sex*race HRV ANOVAs.  Thus, the total degrees of freedom for the design is 139 (n-1), and since 2x2 ANOVA with interaction models are fit for each of the HRV endpoints, there are 3 model degrees of freedom (df=1 for sex, df=1 for race, df=1x1=1 for the sex*race interaction), leaving 139-3 = 136 error degrees of freedom.  So each F test for an effect here has 1 and 136 df.  The significant test results reported (HFP, LFP, pNN50) are for the main effect of race.  These are valid main effects to assess since the interaction term in these models were insignificant.

  1. Delete space after minus  power (? _= 0.205, p= -0.214 and after mixed

I could delete the space in the abstract after "mixed" but I could not do so for the other values.  Maybe the editor can help with the formatting.

2. Rewrite this sentence "all athletes reported LF and pNN50 values that were significantly correlated with CVD risk factors: DBP and BW." 

This has been re-written and is a stand-alone sentence with a minor change that I think makes the sentence easier to understand. It is highlighted in the abstract. It reads: All athletes reported LF and pNN50 values that were significantly correlated with two CVD risk factors: DBP and BW. 

3. Introduction: Change 0700 and 1000 h in 07:00 and 10:00. Done

4. Remove explained from Table 2. Heart Rate Variability metrics explained. Change.  “N-N interval”.  With  “RR interval” in table 2 and in the text.  Done.

5. pag 9 How is reckoned or estimate VO2max ? There were actual VO2max tests performed in the two references (21 and 42). That is now indicated in the text.

6. Pag 10 It is not clear: "Although in that study gender was not compared"

That part of the sentence is removed for clarity.

We appreciate this reviewer's attention to detail

Reviewer 4 Report

The paper has been  sufficiently improved . 

Author Response

Appreciate that the reviewer appears satisfied with changes made on first draft.